# SYK kinase mediates brown fat differentiation and activation

Marko Knoll[1], Sally Winther[1,2], Anirudh Natarajan[1], Huan Yang[1], Mengxi Jiang[1], Prathapan Thiru[1], Aliakbar Shahsafaei[3], Tony E. Chavarria[1], Dudley W. Lamming [1,4], Lei Sun[1,5], Jacob B. Hansen [2] & Harvey F. Lodish[1,6]

Brown adipose tissue (BAT) metabolism influences glucose homeostasis and metabolic health in mice and humans. Sympathetic stimulation of β-adrenergic receptors in response to cold induces proliferation, differentiation, and UCP1 expression in pre-adipocytes and mature brown adipocytes. Here we show that spleen tyrosine kinase (SYK) is upregulated during brown adipocyte differentiation and activated by β-adrenergic stimulation. Deletion or inhibition of SYK, a kinase known for its essential roles in the immune system, blocks brown and white pre-adipocyte proliferation and differentiation in vitro, and results in diminished expression of *Ucp1* and other genes regulating brown adipocyte function in response to β-adrenergic stimulation. Adipocyte-specific SYK deletion in mice reduces BAT mass and BAT that developed consisted of SYK-expressing brown adipocytes that had escaped homozygous *Syk* deletion. SYK inhibition in vivo represses β-agonist-induced thermogenesis and oxygen consumption. These results establish SYK as an essential mediator of brown fat formation and function.

[1] Whitehead Institute for Biomedical Research, 455 Main Street, Cambridge, MA 02142, USA. [2] Department of Biology, University of Copenhagen, Universitetsparken 13, DK-2100 Copenhagen, Denmark. [3] Department of Pathology, Brigham and Women's Hospital, 75 Francis Street, Boston, MA 02115, USA. [4] Department of Medicine, University of Wisconsin-Madison, 1685 Highland Avenue, Madison, WI 53705, USA. [5] Cardiovascular and Metabolic Disorders, Duke-NUS Graduate Medical School, 8 College Road, Singapore 169857, Singapore. [6] Departments of Biology and Biological Engineering, Massachusetts Institute of Technology, 21 Ames Street, Cambridge, MA 02142, USA. Marko Knoll and Sally Winther contributed equally to this work. Correspondence and requests for materials should be addressed to H.F.L. (email: lodish@wi.mit.edu)

Adipose tissue is an essential regulator of energy balance and nutritional homeostasis[1]. Of the two principal types of adipose tissue, white adipose tissue (WAT) is specialized to store chemical energy in the form of triglycerides. On the other hand, brown adipose tissue (BAT), is specialized to generate heat and consume energy as a defense against cold; BAT protects from excessive weight gain in response to overfeeding[1, 2]. Cold exposure, through β-adrenergic signaling, induces changes in both tissues. In BAT, cold exposure increases Ucp1 expression, mitochondrial biogenesis, and tissue expansion, resulting in an increase of thermogenic capacity[2]. In WAT, cold exposure leads to the emergence of beige, or inducible brown, adipocytes[3].

Brown adipocyte differentiation and activation in response to β-adrenergic stimulation can be modeled in vitro by inducing ex vivo isolated brown adipocyte precursors or precursor cell lines to assume a brown adipocyte-like phenotype, followed by stimulation with β-adrenergic agonists such as isoproterenol[4–6]. β-adrenergic stimulation, by cold or pharmacological agents, initiates numerous processes in the brown adipocytes aimed at increasing thermogenic activity and tissue recruitment. Rapid effects include increased nutrient uptake, lipolytic activation, and activation of UCP1. Prolonged β-adrenergic stimulation induces proliferation and differentiation of brown adipocyte precursor cells, mitochondrial biogenesis, as well as changes in gene expression and browning of WAT to increase thermogenic capacity[7, 8].

Spleen tyrosine kinase (SYK) is critical for survival, differentiation, and activation of several types of hematopoietic cells[9]. Recruitment of its paired SH2 domains to dually phosphorylated tyrosine binding motifs, termed immunoreceptor tyrosine based activation motifs (ITAMs), localizes SYK to ITAM linked receptors such as the B cell receptor (BCR) and activation of SYK is also mediated by Src-family tyrosine kinases such as Lck/Yes novel tyrosine kinase (LYN). Thus, SYK connects the BCR and other immune receptors with downstream activation of numerous pathways resulting in calcium release and transcriptional responses[9].

Although initially characterized as a hematopoietic cell-specific kinase essential for immune receptor signaling, SYK has been ascribed functions in other signaling cascades not only in immune cells but also in cells such as fibroblasts and endothelial cells[10].

Using a kinase inhibitor library, we show here that SYK is required for β-adrenergic stimulated upregulation of Ucp1 in mature brown adipocytes. We utilized numerous inhibitors, gene knock downs, and gene knockout strategies to show that SYK regulates activation, proliferation, and differentiation of brown adipocytes. In vivo, knockout of SYK is incompatible with brown adipose formation, and we observed a strong counter selection for SYK proficient cells. SYK inhibition in established BAT in vivo inhibited β-agonist-induced thermogenesis and oxygen consumption. These results establish SYK as an essential mediator of brown fat formation and function, and suggest that pharmacological modulation of SYK activity could have an impact on certain metabolic diseases.

## Results

**SYK is expressed in BAT and induced during brown adipogenesis.** In order to identify kinases required for isoproterenol-induced Ucp1 expression in brown adipocytes, we pretreated mature immortalized brown adipocytes on day 8 of differentiation with a library of kinase inhibitors (Supplementary Table 1) followed by isoproterenol stimulation, and measured Ucp1 mRNA expression 6 h later. Apart from two pan-kinase inhibitors included in the screen, SYK inhibitor ER27319 (SYK-i1)[11] was

the only compound tested that resulted in almost complete loss of isoproterenol-induced Ucp1 expression (Fig. 1a). Although known mainly for their critical roles in hematopoietic cells, SYK as well as several SYK interaction partners and targets including tyrosine-protein phosphatase non-receptor type 6 (SHP1), Bruton's tyrosine kinase (BTK) and 1-phosphatidylinositol 4,5-bisphosphate phosphodiesterase gamma-2 (PLCγ2) were indeed induced during in vitro brown fat differentiation (Fig. 1b, Supplementary Fig. 5) and detected in whole mature BAT, albeit expression as a fraction of total protein was much lower than in B cells (Fig. 1c, Supplementary Fig. 5). SYK activity is controlled by phosphorylation[9]. Although we were unable to detect phosphorylated SYK by western blotting, we detected an increase of phosphorylated SYK following isoproterenol treatment using immunofluorescence (IF) that was blocked when the cells were pretreated with SYK inhibitor R406 (SYK-i3)[12] (Fig. 1d, Supplementary Fig. 1a) ($n = 5$). Similarly, no phosphorylated SYK immunofluorescence signal was detected when mature day 8 adipocytes deleted for SYK (Fig. 2a) were stimulated with isoproterenol (Supplementary Fig. 1a) ($n = 3$). This indicates that isoproterenol enhances SYK activity by triggering its phosphorylation.

**SYK inhibits β-agonist-induced Ucp1 expression and $O_2$ consumption.** To confirm the identification of SYK in adipocytes as essential for isoproterenol-induced Ucp1 expression in vitro, we used multiple approaches ranging from siRNA knock down, 4 different SYK inhibitors, and Cre-mediated deletion of Syk. We blocked SYK activity using two inhibitors and depleted Syk RNA by ~90% using two different siRNAs (Supplementary Fig. 1b) in immortalized day 8 differentiated brown adipocytes[13] (Fig. 1e). Furthermore, we tested SYK-i3 on primary day 8 differentiated brown adipocytes (Fig. 1f), and cells derived from CreERT2 Syk[flox/flox] (KO) mice or Syk[flox/flox] (WT) control mice in which genetic Syk deletion was induced in vitro by adding 4-hydroxy tamoxifen (4-OHT) with up to 90% deletion in mature brown adipocytes (Fig. 1f and Fig. 2a). In all cases, isoproterenol-induced Ucp1 RNA expression was reduced (Fig. 1e–f). Basal levels of Ucp1 expression in unstimulated cells, whether treated with inhibitors, siRNAs, or in Syk KO cells, were not affected by SYK modulation (Supplementary Fig. 1c–d). Cells pretreated with SYK-i1 at day 8 of differentiation followed by stimulation with isoproterenol for 48 h also failed to induce UCP1 expression at the protein level (Supplementary Fig. 1e, Supplementary Fig. 5). Conversely, overexpression of SYK (Syk-OE) or phosphatase inhibition (Phos-i), which augments SYK activity[14, 15], enhanced Ucp1 expression in isoproterenol treated adipocytes (Fig. 1g, Supplementary Fig. 1f–h, Supplementary Fig. 5), while basal levels of Ucp1 were not affected (Supplementary Fig. 1h). Consistent with the results of isoproterenol stimulation, forskolin, a direct activator of cAMP synthesis, induced Ucp1 upregulation in WT but not KO cells (Supplementary Fig. 1i).

Further, isoproterenol-mediated acute oxygen consumption in mature brown adipocytes was impaired by pretreatment with the SYK inhibitor Bay61-3606 (SYK-i2)[16] (Fig. 1h, left panel). Knock down of Syk starting at day 6 of differentiation already lowered the basal oxygen consumption at day 10 of differentiation and treated cells were also impaired in isoproterenol-mediated acute induction of oxygen consumption (Fig. 1h, right panel). Finally, we showed that SYK is important for isoproterenol-induced lipolysis. Glycerol released to the media by brown adipocytes was decreased when the cells were pretreated with SYK-i1 or SYK-i2 (Fig. 1i). Similarly, glycerol release in white adipocytes was also reduced in the presence of SYK inhibitors (Supplementary Fig. 1j).

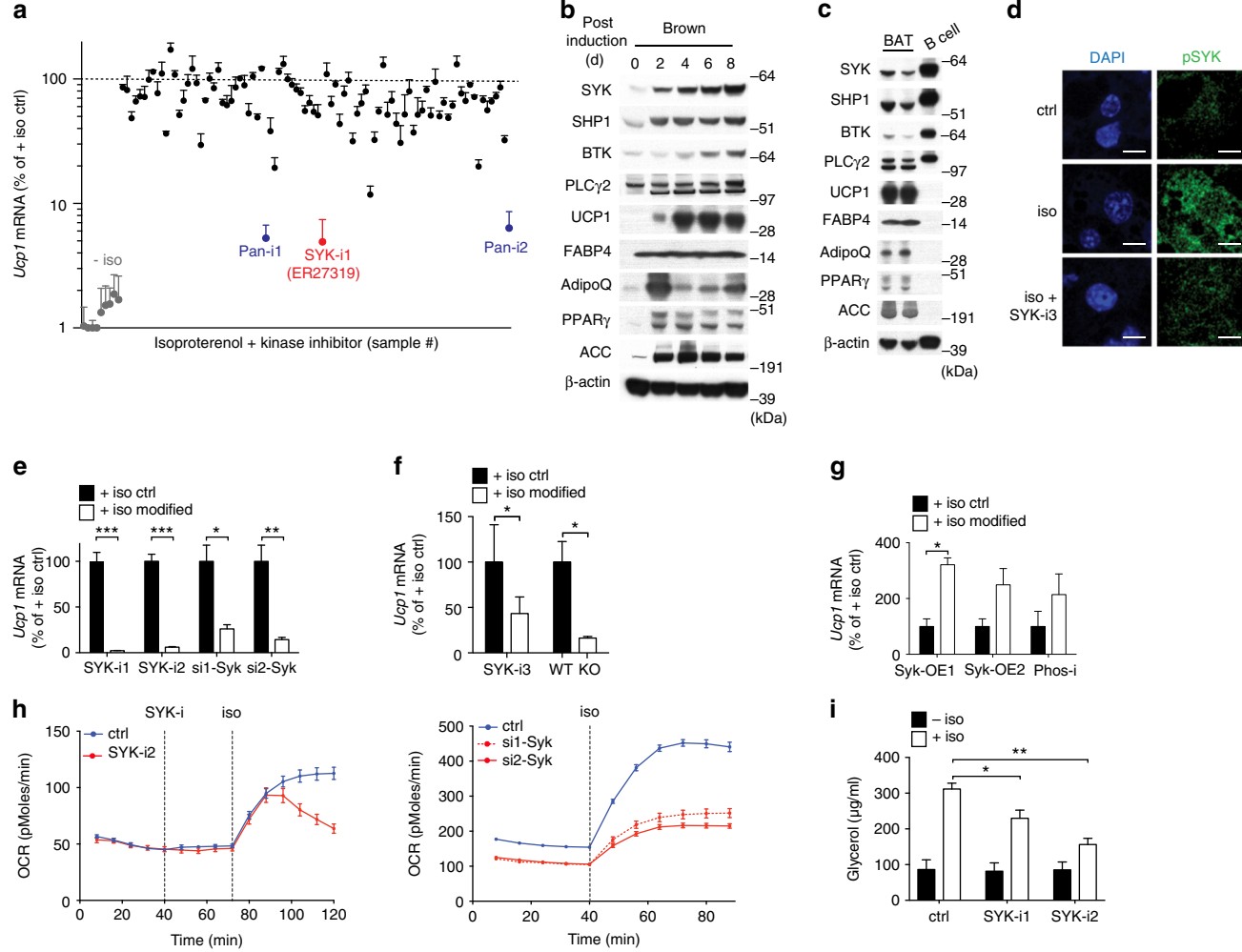

**Fig. 1** Identification of kinases regulating *Ucp1* expression in response to β-adrenergic stimulation. **a** Screening of a kinase inhibitor library (*n* = 90) to identify small molecules modulating *Ucp1* RNA expression in response to stimulation with isoproterenol in in vitro differentiated day 8 immortalized brown adipocytes. Blue, pan-kinase inhibitors; red, SYK inhibitor ER27319; dotted line denotes + isoproterenol control (*n* = 3). **b** Representative immunoblots of two independent in vitro differentiations of primary brown adipocytes (**b**) and of two independent BAT isolations with two mice each (**c**) of SYK and pathway members as well as brown adipocyte markers. **d** Immunofluorescence of primary brown adipocytes stimulated with isoproterenol in the absence or presence of 2 μM SYK-i3 (R406), one representative IF of five independent replicates is shown. Scale bar = 10 μm. **e** *Ucp1* RNA expression after stimulation with isoproterenol of differentiated day 8 immortalized brown adipocytes pretreated with 10 μM SYK inhibitors (SYK-i1 is ER27319; SYK-i2 is Bay 61-3606) or following transfection with siRNA targeting *Syk* mRNA 4 days prior (si1-Syk and si2-Syk) (*n* = 3). **f** Isoproterenol-induced *Ucp1* RNA expression in primary day 8 brown adipocytes in vitro pretreated with SYK-i3 (R406) or in primary day 8 brown adipocytes derived from CreERT2 Syk^flox/flox mice (KO) or Syk^flox/flox (WT) both treated with 4-hydroxy tamoxifen (4-OHT) one day prior to induction of differentiation (*n* = 3). **g** *Ucp1* RNA expression in primary day 8 brown adipocytes after overexpression of wild-type mouse SYK (Syk-OE1 and Syk-OE2) or pretreatment with 1 μM phosphatase inhibitor 3AC and stimulation with isoproterenol (*n* = 2). **h** Oxygen consumption rate (OCR) of immortalized brown adipocytes stimulated with isoproterenol and pretreated with 10 μM (SYK-i2) or following siRNA mediated knock down of *Syk* 4 days prior to measurement (si1-Syk, si2-Syk). Results are shown as average OCR + SEM of 18 measurements per time point and representative of three replicates. **i** Cell culture medium glycerol content from immortalized brown adipocytes at day 8 treated with isoproterenol in the absence or presence of 10 μM SYK-i2 (*n* = 3). For **a**, **e–g**, mean % of iso stimulated control ± SEM. For **a**, **g**, **i**, two-tailed unpaired Student's *t* test was performed (**P* < 0.05, ***P* < 0.005 and ****P* < 0.0005). For **f**, two-tailed unpaired Student's *t* test was performed on Log transformed values

Thus, SYK is expressed in brown fat cells, is developmentally upregulated, and critically mediates the upregulation of *Ucp1*, oxidative metabolism, and lipolysis in response to β-adrenergic stimulation of brown adipocytes.

**SYK mediates global changes in β-agonist-induced gene expression.** To test whether SYK is required for global changes in gene expression in response to β-adrenergic stimulation of brown adipocytes[7], we used cells derived from CreERT2 Syk^flox/flox (KO) or control Syk^flox/flox (WT) control mice and treated them with

4-OHT one day prior to the induction of differentiation; we achieved a 90% reduction in SYK protein at day 8 of differentiation (Fig. 2a, Supplementary Fig. 6). Mature adipocytes were then stimulated with isoproterenol for 6 h and RNA-seq was performed. Unstimulated WT and KO cells were viable, morphologically normal, and displayed largely normal patterns of gene expression as indicated by RNA-seq and qPCR analysis (Supplementary Fig. 2a–b). As seen in Fig. 2b we identified 4372 genes that were at least 1.5-fold up or downregulated with an FDR < 0.05 (gray dots) in response to β-adrenergic stimulation in either the mature WT or KO adipocytes, with most genes

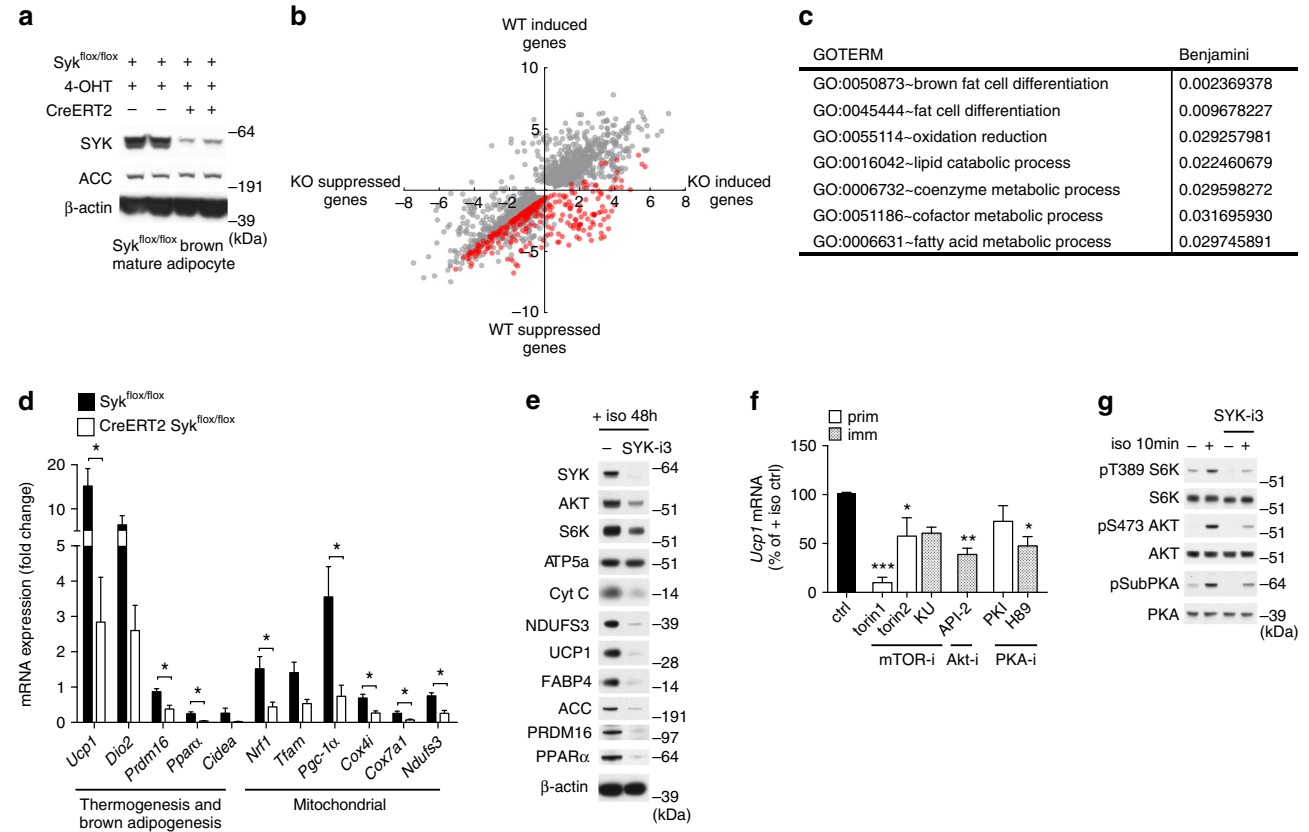

**Fig. 2** SYK mediates expression of characteristic brown fat genes and activates the AKT/mTOR pathway. **a** Representative immunoblots of SYK protein expression from three independent differentiations of in vitro differentiated brown adipocytes from CreERT2 Syk^flox/flox mice on day 8 of differentiation following continuous treatment with 4-hydroxy tamoxifen (4-OHT) for 9 days. **b** RNA-seq was conducted on 3 replicates of 4-OHT treated Syk^flox/flox (WT) and CreERT2 Syk^flox/flox (KO) cells as shown in **a** before and after isoproterenol treatment at day 8 of differentiation. Log fold change in WT cells before and after induction is shown on the x axis and log fold change in KO cells before and after induction is shown on the y axis. Genes shown had at least a 1.5-fold up- or down-regulation in WT or KO cells after stimulation with an FDR < 0.05. Genes with a fold change difference of 1.5 higher in WT compared to KO with a p value < 0.05 are shown in red. **c** Enriched GO categories with a Benjamini adjusted p value < 0.05 for genes shown in red in **b**. **d** Fold change of mRNA expression in day 8 differentiated primary brown adipocytes with tamoxifen-induced reduction in SYK protein as described in (**a**) and stimulated with isoproterenol. **e** Representative immunoblots of two independent experiments of primary brown adipocytes stimulated with isoproterenol on day 8 of differentiation for 48 h in complete medium in the presence or absence of 5 μM SYK-i3. **f** Day 8 differentiated brown adipocytes (prim, primary; imm, immortalized) stimulated with isoproterenol and pretreated with mTORC inhibitors torin 1 (10 μM), torin 2 (10 μM) or KU0063794 (10 μM), AKT inhibitor API-2 (10 μM), or PKA inhibitors PKI 14-22 (10 μM), or H89 (10 μM) (n = 3). **g** Representative immunoblots of three independent experiments of day 8 primary brown adipocytes pretreated with 5 μM SYK-i3 and stimulated with isoproterenol for 5 min. pPKAsub, antibody recognizing PKA substrate phosphorylation motif RRXS*/T*. For **f**, mean % ± SEM and **d**, fold change vs unstimulated control ± SEM was calculated. For **d**, two-tailed unpaired Student's t test was performed (*P < 0.05, **P < 0.005 and ***P < 0.0005) and for **f**, a one-way ANOVA was performed

showing a similar pattern of up- or down-regulation. To identify the genes affected by SYK deletion, we looked for genes that showed a differential fold change >1.5 with a p value < 0.05 upon β-adrenergic stimulation in WT compared to SYK-deleted cells (Methods section). We identified 307 genes, several of which were enriched in pathways involved in brown fat cell differentiation, fat cell differentiation, oxidation-reduction, and others (Fig. 2c, Supplementary Fig. 2c–d). We validated the expression of several genes by qPCR and included others that are important in brown fat cell differentiation and mitochondrial biogenesis. Indeed, *Syk* KO adipocytes, compared to WT adipocytes, showed impaired induction or suppression of many genes characteristic of or required for brown adipogenesis (*Ucp1, Dio2, Prdm16, Pparα,* and *Cidea*) and mitochondrial biogenesis and function following treatment with isoproterenol (*Nrf1, Tfam, Pgc-1α, Cox4i, Cox7a1,* and *Ndufs3*) (Fig. 2d). Stimulation of brown adipocytes that were pretreated with SYK-i3 and subsequently stimulated with iso-proterenol for 6 or 48 h similarly resulted in impaired induction and expression of genes encoding proteins enriched in brown fat,

mitochondria, and adipogenesis at both mRNA (Supplementary Fig. 2e) and protein levels (Fig. 2e, Supplementary Fig. 6), whereas expression of other genes such as those encoding β-actin or an ATP synthase subunit (ATP5a) were normal (Fig. 2e).

To identify downstream targets of SYK, we reanalyzed the screen of kinase inhibitors used in the *Ucp1* expression screen (Fig. 1a) and identified as the next top hits inhibitors of the PI3K, mTOR, AKT, and PKA pathways (Supplementary Fig. 2f). Indeed, inhibition of members of the mTOR, AKT, and PKA pathways, suppressed isoproterenol-induced *Ucp1* expression in both primary and immortalized day 8 differentiated brown adipocytes (Fig. 2f). Inhibition of PI3K using inhibitor PI828 showed 50% inhibition of *Ucp1* expression only at a very-high dose (>25 μM), hence, we are uncertain of the specificity of the used inhibitor. Pretreatment with SYK-i3 reduced isoproterenol-induced phosphorylation of mTOR complex1 (mTORC1) and mTORC2 substrate phosphorylation target S6K, AKT phosphorylation, and PKA substrate phosphoryla-tion in both primary brown adipocytes (Fig. 2g, Supplementary Fig. 6) and immortalized mature brown adipocytes (Supplementary

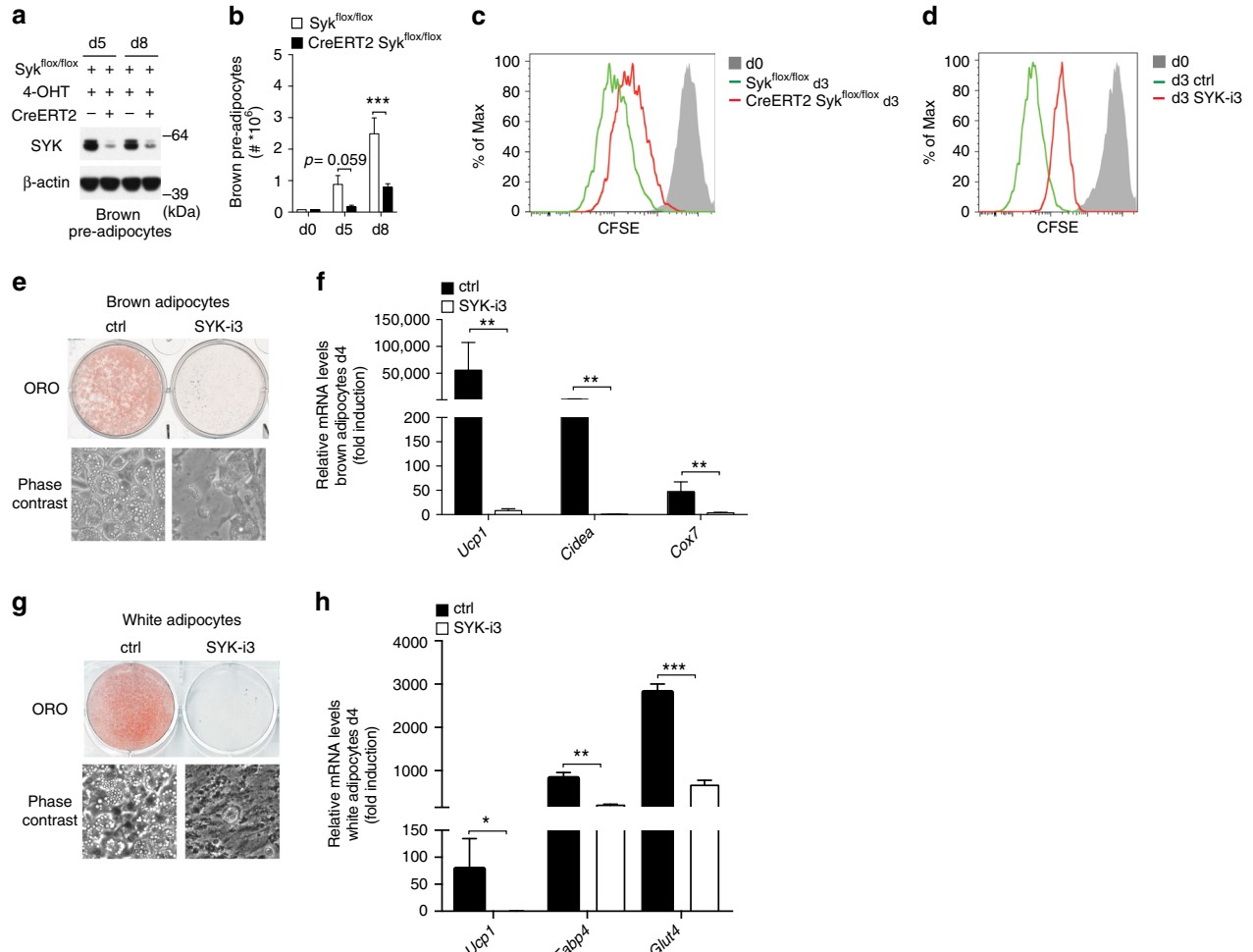

**Fig. 3** SYK is essential for pre-adipocyte proliferation and for adipocyte differentiation. Representative western blot of SYK protein expression (**a**), cell numbers of three independent experiments (**b**) and representative flow cytometry of three independent experiments of CFSE staining analyzed by flow cytometry (**c**) of brown CreERT2 Syk$^{flox/flox}$ and Syk$^{flox/flox}$ pre-adipocytes seeded at equal cell numbers (d0) following continuous 4-hydroxy tamoxifen (4-OHT) treatment beginning 3 days prior. **d** Representative flow cytometry of three independent experiments of primary brown pre-adipocytes 3 days after labeling with CFSE and addition of 5 μM Syk-i3. Oil red oil staining (ORO), phase contrast images, and mRNA expression 4 days post induction of differentiation of primary brown (**e**, **f**) and white (**g**, **h**) pre-adipocytes in the presence of 5 μM SYK-i3 in the induction media; data are representative of three independent replicates. Mean cell number + SEM (**b**) and fold induction vs. d0 + SEM (**f**, **h**). For **b**, two-tailed unpaired Student's $t$ test was performed (*$P < 0.05$, **$P < 0.005$ and ***$P < 0.0005$) and for **f**, **h**, two-tailed unpaired Student's $t$ test was performed on Log transformed data

Figs. 2g and 6). These data show that SYK is likely upstream of these pathways.

In summary, in in vitro cultures of brown adipocytes stimulated by β-adrenergic agonists, SYK promotes and is necessary for expression of numerous genes involved in brown adipogenesis and brown adipocyte mitochondrial biogenesis; SYK is also essential for activation of critical downstream signaling pathways, including the mTOR, AKT and PKA pathways.

**SYK is required for pre-adipocyte proliferation and differentiation.** Acute sympathetic stimulation of BAT leads to acute brown adipocyte activation, and chronic stimulation induces proliferation of brown adipocyte precursors, their differentiation to brown adipocytes, and ultimately tissue growth[7, 17]. The mTOR and PKA pathways critically regulate these processes in vitro and in vivo[18], and since we place SYK upstream of these pathways it raises the possibility that SYK is also required for both brown pre-adipocyte proliferation and brown adipocyte differentiation. To test this hypothesis, we isolated brown and white pre-adipocytes from CreERT2 Syk$^{flox/flox}$ (KO) and control Syk$^{flox/flox}$ (WT) mice and

treated both with 4-OHT. Treatment resulted in a ~80% loss of SYK in CreERT2 Syk$^{flox/flox}$ (KO) but not in Syk$^{flox/flox}$ (WT) control cells (Fig. 3a, Supplementary Fig. 7). Indeed, brown and white pre-adipocyte proliferation required SYK, as indicated by diminished cell numbers (Fig. 3b and Supplementary Fig. 3a). The reduced cell numbers could not be explained by an increase of apoptosis, as staining with Annexin V and 7-AAD did not show a significant increase of apoptotic or dead cells in KO cultures (Supplementary Fig. 3b–d). We then tested cell proliferation in brown pre-adipocytes by CFSE staining and detected delayed cell division both in *Syk* KO cells (Fig. 3c) and SYK-i3-treated cells (Fig. 3d).

For proper induction of differentiation, pre-adipocytes need to be cultured to the point where they are growth arrested. Pre-adipocytes derived from CreERT2 Syk$^{flox/flox}$ mice were delayed in cell growth, did not reach confluency, and did not differentiate to the same extent as did cells from Syk$^{flox/flox}$ control mice (Supplementary Fig. 3e). The few brown adipocytes formed after 8 days expressed normal protein levels of SYK (Supplementary Figs. 3f and 7), suggesting that only the few SYK proficient brown pre-adipocytes that escaped homozygous *Syk* deletion were able to differentiate in these cultures. Thus, since SYK is required for

pre-adipocyte proliferation, these experiments cannot be used to determine whether SYK additionally is required for brown adipocyte differentiation.

To show that SYK is necessary for brown and white adipocyte differentiation, we added two different SYK inhibitors to the induction media of primary wild-type brown and white pre-adipocytes as well as to immortalized brown pre-adipocytes; both inhibitors blocked brown and white adipocyte differentiation without impairing viability (Fig. 3e, g, Supplementary Fig. 3g). In addition, these inhibitors caused reduced expression of genes critical for brown and white fat differentiation and mitochondrial function at day 4 of differentiation (Fig.3f, h, Supplementary Fig. 3h). In summary, SYK is essential for both brown and white pre-adipocyte proliferation and for brown and white adipocyte differentiation in vitro, consistent with a role for SYK signaling in mTOR and PKA activation and expression of genes involved in adipogenesis as well as mitochondrial biogenesis and function. Determining how SYK becomes activated in both pre-adipocytes and adipocytes, and how SYK activates multiple downstream signaling pathways remain to be deciphered.

**SYK is important for generation of brown adipocytes in vivo.** To determine whether SYK is also required for brown adipocyte differentiation in vivo, we generated mice with a fat-specific deletion of the *Syk* gene by combining an adiponectin promoter driven Cre recombinase allele (AdipoQCre) with a *Syk* allele harboring loxP sites flanking exon 1[19, 20]. When housed at 20 °C, only 77% AdipoQCre Syk$^{flox/flox}$ mice were alive at weaning age compared to Syk$^{flox/flox}$ control mice (Fig. 4a); this may account for part of the postnatal mortality observed in mice with a germline deletion of *Syk*[21, 22]. This phenotype was not observed when the mice were housed at 30 °C (Fig. 4a), suggesting a brown fat defect in AdipoQCre Syk$^{flox/flox}$ mice.

Consistent with impaired brown adipocyte precursor expansion in vitro (Fig. 3a–d), BAT depot mass was decreased in four week old AdipoQCre Syk$^{flox/flox}$ mice compared to Syk$^{flox/flox}$ control mice (Fig. 4b). The remaining BAT of AdipoQCre Syk$^{flox/flox}$ mice expressed normal levels of SYK protein and was otherwise similar to controls, as judged by immunoblotting and qPCR of BAT enriched genes, mitochondrial genes, adipogenic genes and genes involved in lipid metabolism (Fig. 4d, Supplementary Figs. 4a and 8). The subcutaneous inguinal white adipose tissue (scWAT) of AdipoQCre Syk$^{flox/flox}$ mice displayed increased expression of BAT enriched genes, compared to scWAT from Syk$^{flox/flox}$ control mice, as well as histological features of browning and UCP1 expression. (Fig. 4c, Supplementary Figs. 4b–d and 9). This suggested compensatory beige adipocyte formation from the remaining progenitor cells that escaped homozygous *Syk* deletion, reminiscent of other mouse models with decreased BAT mass[23]. Similarly, we generated mice with a Ucp1Cre driven brown fat-specific deletion[24] and bearing a *Syk* allele harboring loxP sites (Ucp1Cre Syk$^{flox/flox}$). The BAT in these mice also expressed normal levels of SYK protein (Supplementary Figs. 4e and 9). Fat-specific SYK deletion both in culture and in vivo thus results in decreased BAT mass, favoring the proliferation and differentiation of residual SYK-expressing brown adipocytes.

As expected from the reduced amount of apparently normal BAT that escaped Syk deletion, AdipoQCre Syk$^{flox/flox}$ mice, on a high fat diet (HFD) gained more weight and became more glucose intolerant than Syk$^{flox/flox}$ control mice. They also displayed insulin resistance on a chow diet (Fig. 4e–f, Supplementary Fig. 4f–g). RNA expression of genes regulating lipolysis and lipogenesis in scWAT, visWAT and BAT of these mice was similar to controls, suggesting that obesity in these mice may not be due to a lipolysis or lipogenesis defect (Supplementary Fig. 4h).

To confirm the deletion of the *Syk* gene in BAT of AdipoQCre Syk$^{flox/flox}$ mice, we isolated pre-adipocytes from BAT depots of AdipoQCre Syk$^{flox/flox}$ and Syk$^{flox/flox}$ control mice as well from heterozygous Syk$^{+/flox}$ and AdipoQCre Syk$^{+/flox}$ mice and induced them to differentiate in vitro. We were able to detect faint amounts of the deleted allele at day 0, before induction of differentiation in progenitors from the AdipoQCre Syk$^{flox/flox}$ and AdipoQCre Syk$^{+/flox}$ mice but not the control mice, indicating a low level of expression of the AdipoQCre transgene and Cre activity in the progenitors (Fig. 4g, Supplementary Fig. 8). During differentiation of these AdipoQCre expressing progenitor cells in culture the intensity of the deleted allele "band" increased, correlating with induction of AdipoQCre expression during BAT differentiation (Fig. 4g). Thus the absence of deleted Syk in mature BAT cells from AdipoQCre Syk$^{flox/flox}$ mice is not due to faulty Cre recombinase activity, but rather to poor differentiation of Syk-deleted BAT progenitors. By quantifying the extent of the deletion by Southern blotting we found a ~50 % deletion of the loxP-flanked *Syk* allele in BAT of AdipoQCre Syk$^{flox/flox}$ as well as in Ucp1Cre Syk$^{flox/flox}$ mice, compared to heterozygous germline deleted BAT (Fig. 4h, Supplementary Figs. 4i, 8 and 9). This again suggested selection for brown adipocytes with a heterozygous (or no) *Syk* deletion and against those with complete *Syk* deletion. We also detected the deleted allele in scWAT and visWAT of AdipoQCre Syk$^{flox/flox}$ mice (Supplementary Figs. 4j and 9).

Consistent with a requirement for SYK in brown pre-adipocyte proliferation (Fig. 3a–d), these results suggest that Cre-mediated deletion of SYK beginning in proliferating brown adipocyte precursors selects for the development of fewer but otherwise normal SYK-expressing brown adipocytes that have escaped homozygous deletion of the loxP-flanked *Syk* allele. Thus, SYK deficiency is incompatible with brown adipocyte development, indicating that SYK is essential for brown fat formation in vivo.

**SYK inhibition impairs thermogenesis in vivo.** To address whether SYK function is important in established brown fat, we injected mice with a highly selective SYK inhibitor PRT062607, one that is longer-lasting in vivo than the SYK inhibitors we used in our cell culture experiments. We measured oxygen consumption before and after stimulation of β-adrenergic signaling in brown fat by the β3-agonist CL 316,243[25]. Mice injected with the vehicle and subsequent injection of CL 316,243 increased oxygen consumption whereas mice preinjected with SYK inhibitor PRT062607 failed to induce CL 316,243-dependent oxygen consumption (Fig. 4i, upper panel) in the absence of any significant changes in physical movements (Fig. 4i, lower panel). Moreover, mice injected with PRT062607, but not controls, underwent hypothermia upon cold exposure (Fig. 4j). We then deleted SYK in established BAT using CreERT2 Syk$^{flox/flox}$ mice or Syk$^{flox/flox}$ control mice by injecting tamoxifen intraperitoneally (i.p.) for 5 days, which resulted in an almost complete loss of SYK protein 16 days post tamoxifen injection (Fig. 4k, Supplementary Fig. 8). This SYK deficient BAT was grossly normal morphologically and displayed normal expression of UCP1 and other mitochondrial and adipogenesis enriched genes, as well as of PKA, S6K, and AKT (Fig. 4k, Supplementary Figs. 4k, 8 and 9). SYK is thus not required for survival of established BAT or maintenance of expression of brown fat characteristic genes. However, as judged by measuring the oxygen consumption of these mice in metabolic chambers, CreERT2 Syk$^{flox/flox}$ mice failed to induce CL 316,243-induced oxygen consumption, as contrasted with control Syk$^{flox/flox}$ mice (Fig. 4l). SYK signaling thus promotes β-adrenergic induced BAT activation in vivo.

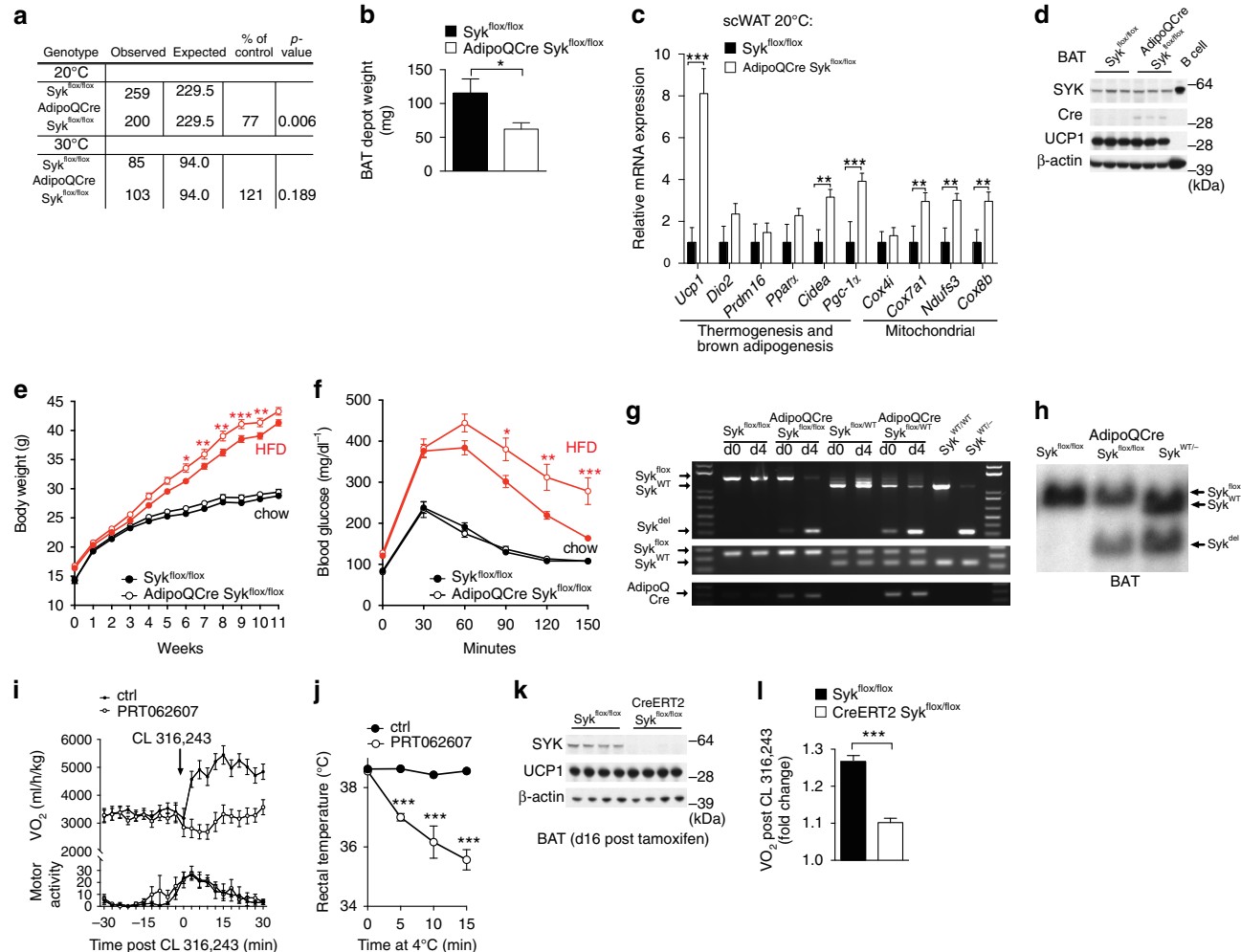

**Fig. 4** SYK deficiency is incompatible with brown fat formation and activation. Genotypes of offspring at weaning age from AdipoQCre Syk[flox/flox] mice mated with Syk[flox/flox] mice (**a**). Weight of complete BAT depots from 4 week old mice ($n = 5$ each genotype) on a chow diet (**b**). **c** RNA expression in subcutaneous white adipose tissue scWAT (AdipoQCre Syk[flox/flox], $n = 5$; Syk[flox/flox], $n = 4$) and representative immunoblots (**d**) of BAT ($n = 3$) from 7–8 week old mice on a chow diet housed at 20 °C. Body weights (**e**) and glucose tolerance test (**f**) of AdipoQCre Syk[flox/flox] mice or Syk[flox/flox] mice on a high fat diet (HFD) in red (AdipoQCre Syk[flox/flox], $n = 17$; Syk[flox/flox] $n = 18$,) and chow diet (chow) in black (AdipoQCre Syk[flox/flox], $n = 15$; Syk[flox/flox], $n = 21$). Asterisks shown are multiple hypothesis corrected $p$ values < 0.05 for an unpaired two-tailed $t$ test of weights (**e**) or blood glucose (**f**) per time point. PCR amplification (**g**) of the deleted Syk[flox] allele (top), Syk[flox] genotype (middle) and the AdipoQCre allele (bottom) in brown pre-adipocytes (d0) and differentiated brown adipocytes (d4) derived from BAT of the indicated genotype and heterozygous germline deleted mice. **h** Southern blot of BamHI digested genomic DNA of BAT from 6 weeks old mice with the indicated genotypes. **i** Oxygen consumption (upper panel) and motor activity (lower panel) before and after intraperitoneal (i.p.) injection of CL316,243 with ($n = 13$, female mice) or without SYK-i4 (PRT 062607) ($n = 13$, female mice). **j** Core body temperature after acute cold exposure with or without i.p. injected SYK inhibitor PRT 062607 ($n = 5$). **k** Immunoblotting of BAT from 10 week old CreERT2 Syk[flox/flox] mice and Syk[flox/flox] controls ($n = 4$, male mice) 16 days after last tamoxifen treatment. **l** Oxygen consumption measured (VO₂) during the sleep cycle was compared with VO₂ measured for 12 h post injection with CL316,243 subcutaneous (s.c.) of the mice shown in **k** ($n = 4$). For **i**–**j**, **l** $P$ values were calculated with the two-way ANOVA test. For **a**, $P$ values were determined using a two-tailed chi-square test, (**b**) data are shown as mean ± SEM, asterisks denote statistical significance ($p < 0.05$) determined by two-tailed $t$ test and for **c**, asterisks denote statistical significance ($p < 0.05$) determined by two-tailed $t$ test

## Discussion

This study revealed an unexpected and crucial role for the immune kinase SYK in brown adipocyte differentiation, activation, and function. In mature brown adipocytes in vitro and in vivo, inhibition or deletion of SYK blocked upregulation of *Ucp1* expression and oxygen consumption in response to β-adrenergic signaling. In hematopoietic cells, SYK activation requires binding to phosphorylated ITAMs associated with immunoreceptors, and subsequent phosphorylation of SYK by LYN family protein tyrosine kinases[9]. Indeed we detected SYK phosphorylation in response to β-adrenergic stimulation but the precise molecular mechanism of SYK activation in brown

adipocytes is unknown. It may require the association with a yet unidentified YXXL/I ITAM motif in a BAT protein. Alternatively, a yet unknown ITAM-independent pathway could be involved.

In line with our data, other studies have also shown that β-adrenergic stimulation of brown adipocytes induces mTOR signaling measured by increased Akt Ser473 and S6K phosphorylation[26, 27]. Although we found that SYK enhances mTOR activation, measured as increased phosphorylation of the downstream effector S6K, we have been unable to identify proteins directly phosphorylated by SYK. This is a major ongoing effort in our lab. This determination is not straightforward as we do not know which are direct and indirect targets of this kinase and

interactions of kinases and target proteins are usually very transient.

Most importantly, we found that adipose- or BAT-specific deletion of SYK, using AdipoQCre or Ucp1Cre driven inducers of deletion in vivo, probably starting early after birth, results in strong counter selection of SYK proficient cells. This phenomenon has previously been observed in B cells with conditional gene targeting of essential genes[28, 29]. As a result of this counter selection, the mice display reduced BAT mass and predisposition to weight gain and glucose intolerance. This phenotype resembles that of other knockout models with reduced BAT mass[30, 31].

SYK deletion or inhibition decreased proliferation and differentiation of brown adipocyte precursors in vitro and prevented normal BAT development. However, once brown adipocytes have formed, SYK was dispensable in maintaining basal functions of mature brown fat cells but was still necessary for the thermogenic response to β-adrenergic signaling.

A better understanding of the regulation of SYK activity and its targets in brown fat and possibly also in white fat thus has the potential to reveal novel avenues towards the treatment of metabolic diseases.

## Methods

**Mice.** C57BL/6J mice were bred in house or purchased from Jackson Laboratories (stock #000664). Syk$^{flox/flox}$ mice backcrossed 8 times to C57BL/6 mice were a kind gift of Alexander Tarakhovsky, Rockefeller University, and can be purchased from Jackson Laboratories (stock # 017309)[19]. To generate mice with an inducible Syk deletion, Syk$^{flox/flox}$ mice were crossed with UBC-CreERT2 mice (Jackson laboratories #008085, CreERT2 in the manuscript). AdipoQCre mice on a C57BL/6 background were a gift of Evan Rosen, Beth Israel Deaconess Medical Center, and are available at Jackson Laboratories (stock #010803). Ucp1Cre mice were obtained from Jackson laboratories (stock #024670)[24]. In all experiments, Syk$^{flox/flox}$ mice bearing a Cre allele were compared to sex-matched Syk$^{flox/flox}$ littermate or age-matched controls. To generate mice with a germline heterozygous Syk deletion, Syk$^{flox/flox}$ mice were bred with CMV-Cre mice (Jackson laboratories # 006054). All mice were housed under a 12 h light/dark cycle at constant temperature (20 °C). All procedures were performed according to protocols approved by the Committee on Animal Care at the Massachusetts Institute of Technology.

**Pre-adipocyte isolation.** 6–8 2-week to 4-week-old male mice were sacrificed by CO$_2$ asphyxiation and interscapular brown adipose tissue (BAT) and subcutaneous (inguinal) white adipose tissue (scWAT) was harvested into room temperature plain DMEM (Sigma, #56499C). Then the fat pads were transferred into a well of a 6 well plate, excessive media removed and minced with scissors for 5 min. Minced tissues were then transferred into a 50 ml conical tube with 3 ml Hank's balanced salt solution (Gibco, #14175–095) supplemented with 0.2% collagenase A (Roche, #10103578001) and 2 % BSA (Sigma-Aldrich, #A7906) using a 1 ml pipet tip with the tip cut off to allow aspiration of larger pieces. The tissues were incubated agitating (350 rpm) and repeated vortexing every 5 min for 10 s at 37 °C for 30 min. Following collagenase digestion, 10 ml room temperature plain DMEM was added and cells were filtered through a 70 μm mesh filter (Corning, #352350). Mature adipocytes and the stromal vascular fraction (SVF) were separated by centrifugation at 700 g for 5 min. The supernatant was removed and the SVF resuspended in 10 ml room temperature plain DMEM followed by additional filtering through a 30 μm mesh filter (Miltenyi Biotec #130-041-407) and subsequent centrifugation at 700 g for 5 min. The SVF from subcutaneous white fat pads (scWAT) of 8 mice were then resuspended in 10 ml proliferation media Advanced DMEM/F12 (DMEM/F12, LifeTechnologies/Gibco #12634028) supplemented with 5% heat inactivated newborn calf serum (HI NCBS, LifeTechnologies/Gibco, #26010-074), 5% fetal calf serum (FBS, Sigma-Aldrich F2442), 100 units penicillin, 100 μg streptomycin (Pen/Strep, LifeTechnologies/Gibco, #15140-122, 100×), 1× Glutamax (LQX, LifeTechnologies/Gibco, #35050-061) and plated on two 10 cm dishes (Corning, #430293) in a total volume of 10 ml per 10 cm plate. The SVF from interscapular brown fat pads of 6–8 mice were then resuspended in 6 ml DMEM/F12 supplemented with 5% HI NCBS, 5% FBS, Pen/Strep and LQX as above and plated on 3 wells of a six well plate (Corning, #3506) in a total volume of 2 ml per well. After 4 and 24 h, the medium was replaced by fresh, pre-warmed DMEM/F12/5% HI NCBS, 5% FBS, Pen/Strep and LQX at 37 °C and with 5% CO$_2$. Cells were grown to confluency and then passaged no more than two times before seeding the pre-adipocytes for differentiation.

**Pre-adipocyte differentiation.** Pre-adipocytes derived from BAT were seeded in experimental plates at a density of $5 \times 10^5$ per six well plate after the second passage and cultured in proliferation media as described above to confluence for 4 days when the cells were growth arrested. The cells were then induced to differentiate by

culturing them for two days in induction medium consisting of DMEM supplemented with 10% fetal bovine serum and 850 nM insulin (Sigma-Aldrich #I1882), 0.5 μM dexamethasone (Sigma-Aldrich #D4902), 250 μM 3-isobutyl-1-methylxanthine (IBMX, Sigma-Aldrich #I5879), 1 μM rosiglitazone (Cayman Chemical #71742), 1 nM 3,3,5-triiodo-L-thyronine (T3, VWR #100567-778) and 125 nM indomethacin (Sigma-Aldrich #I7378). Subsequently, after two days the induction medium was replaced with DMEM supplemented with 10% FBS and 160 nM insulin and 1 nM T3 for another 2 days. The cells were then cultured in DMEM 10% FBS and 1 nM T3 until day 8 of differentiation, and the medium was replaced every other day. Pre-adipocytes derived from WAT were seeded, grown and differentiated in the same way as pre-adipocytes derived from BAT but no T3 or indomethacin was added to the induction media.

**Differentiation of immortalized brown adipocyte precursors.** The immortalized brown pre-adipocyte cell line (WT-1) was previously established by transformation of brown pre-adipocytes from wild-type newborn mice with SV40 large T antigen and kindly provided by Dr. C. Ronald Kahn[10]. These mycoplasma free cells were propagated in DMEM containing 10% FBS, 62.5 μg/ml penicillin and 100 μg/ml streptomycin (all from Life Technologies). Two days post confluence (designated day 0) differentiation was induced by addition of propagation medium supplemented with 1 μM dexamethasone (Sigma-Aldrich), 0.5 mM IBMX (Sigma-Aldrich), 5 μg/ml insulin (Roche) and 0.5 μM rosiglitazone (Cayman Chemical). On day 2, the medium was replaced with medium containing 5 μg/ml insulin and 0.5 μM Rosiglitazone. Beginning day 4, the cells were cultured in propagation medium until analysis on day 8 or day 10 of differentiation.

**Adipocyte stimulation with isoproterenol.** Mature adipocyte cell layers were washed twice in plain pre-warmed DMEM and cultured in plain DMEM with inhibitors prior to stimulation with the following isoproterenol (Sigma-Aldrich I6504) concentrations: 0.1 μM immortalized day 8 and day 10 brown adipocytes; 1 and 0.1 μM d8 brown adipocytes. After 6 h of treatment, the cells were washed once with cold PBS and RNA was harvested using TRizol or QIAzol lysis reagent as described below. For immunoblotting, the cultures were harvested after indicated treatment times with isoproterenol after washing with cold PBS on ice and adding 50 μl RIPA buffer per well of a 6 well plate or 100 μl RIPA buffer to a 10 cm dish. For stimulation with forskolin (Sigma-Aldrich F3917) 1μM dissolved in DMSO was used.

**Chemical inhibitors.** All inhibitors were added to adipocyte cultures after addition of plain DMEM and 30 min (primary cultures) to 1 h (immortalized cultures) prior to addition of isoproterenol. SYK-i1/ER27319 (Tocris Bioscience), SYK-i2/BAY61-3606 (Sigma-Aldrich), SYK-i3/R406 (free base, Selleck Chemicals #S1533), SYK-i4/PRT062607 (Selleck Chemicals #S8032) and SHP1 inhibitor 3AC (EMD Millipore #5658351) were reconstituted in DMSO and stored at −80 °C. mTOR inhibitors (Torin 1, torin 2, KU0063794), Akt inhibitor API-2, PKA inhibitor PKI 14-22 were all obtained from Tocris Bioscience, PKA inhibitor H89 was purchased from Santa Cruz Biotechnology were all reconstituted in DMSO, stored at −80 °C and used at a final concentration of 10 μM.

**Kinase inhibitor screen.** Different kinase inhibitors ($n = 90$) were added directly to the cell culture well 1 h before stimulation with 0.1 μM isoproterenol for 6 h. The experiment was performed in 12-well plates and each plate contained 10 wells stimulated with isoproterenol and pretreated with 10 μM kinase inhibitor and two control wells, one unstimulated and one isoproterenol stimulated well. Relative Ucp1 RNA expression was normalized to the plate specific isoproterenol stimulated control for each individual experiment. An overview of kinase inhibitors included in the screen is provided in Supplementary Table 1.

**Induction Of Cre recombinase by 4-hydroxy tamoxifen (4-OHT) in culture.** To generate mature, day 8 differentiated Syk KO cells, brown pre-adipocytes were isolated from CreERT2 Syk$^{flox/flox}$ male mice or littermate Syk$^{flox/flox}$ control male mice, grown to confluency in proliferation media and 1 μM 4-hydroxy tamoxifen (4-OHT, Sigma-Aldrich #7904; reconstituted in ethanol) was added 1 day prior to induction of differentiation (d-1). The next day, cells were induced to differentiate (d0) as described above in the presence of 1 μM 4-OHT and complete medium change every other day with fresh 1 μM 4-OHT for 8 days. These in vitro differentiated day 8 brown adipocytes derived from CreERT2 Syk$^{flox/flox}$ mice or Syk$^{flox/flox}$ control mice were washed twice in plain DMEM and starved for one hour prior to stimulation with 0.1 μM isoproterenol or 1 μM forskolin.

To delete Syk in brown pre-adipocytes, again, cells were isolated from CreERT2 Syk$^{flox/flox}$ male mice or littermate Syk$^{flox/flox}$ control male mice cultivated in proliferation media for 3 days, passaged the first time and seeded with the same cell number in the presence of 1 μM 4-OHT. After another 3 days, cells were passaged again and seeded with the same cell numbers counting as day 0 in Fig. 3b although there was no statistical difference in cell numbers at that point. A representative plate was harvested and counted after 5 and 8 days in culture as shown in Fig. 3b or for western at indicated timepoints as seen in Supplementary Fig. 3f 1 μM 4-OHT was added freshly every other day with a complete medium change. After the last counting at day 8, cells were induced to differentiate as described above.

**Apoptosis assay.** Brown or white pre-adipocytes from CreERT2 Syk$^{flox/flox}$ male mice or littermate Syk$^{flox/flox}$ control male mice were treated as described above. After counting, $2 \times 10^5$ cells were used to stain with Annexin V and 7-AAD using the Violet Annexin V/Dead Cell Apoptosis Kit (Life Technologies, A35136) following the instructions of the manufacturer. Briefly, $2 \times 10^5$ cells were resuspended in 200 µl Annexin V binding buffer and 5 µl of Pacific Blue annexin V and 1 µl of SYTOX AADvanced Dead Cell Stain (500 µM) were added. The cells were incubated at room temperature for 30 min, washed twice in Annexin V binding buffer and analyzed using an LSR II flow cytometer.

**CFSE staining.** Brown pre-adipocytes from CreERT2 Syk$^{flox/flox}$ male mice or littermate Syk$^{flox/flox}$ control male mice were treated as described above. After 4 days in the presence of 1 µM 4-OHT, cells were washed twice in pre-warmed plain DMEM media and incubated with 1 ml DMEM supplemented with 1 µl CellTrace stock solution (5 mM) at 37 °C for 20 min. Cells were then washed twice in DMEM with 10% FBS to remove residual free dye and incubated with DMEM in the presence of 10% FBS and 1 µM 4-OHT. Cells were harvested every day for 5 days and CFSE dilution was measured using a LSRII flow cytometer. Wild-type brown pre-adipocytes were stained similar but after the initial staining (day 0) cells were incubated in the presence of SYK-i3 (R406) and measured every day for 5 days.

**Syk overexpression.** Brown pre-adipocytes were transduced with either of two retroviral vectors expressing wild-type mouse Syk tagged with mCherry[32] one day prior to differentiation (d-1) for use on day 8 of differentiation.

**siRNA transfection.** Was performed as described in Isidor et al.[13] Briefly, on day 6 of differentiation, immortalized brown pre-adipocytes were transfected with 5 nM siRNA (Sigma-Aldrich) using 5 µl/ml Lipofectamine RNAiMAX diluted in Opti-MEM I Reduced Serum Medium (Life Technologies). The cultures were analyzed on day 10 of differentiation.

**In vitro oxygen consumption rate.** Measurement of cellular oxygen consumption rate (OCR) was performed using the Seahorse XF96 Extracellular Flux Analyzer (Seahorse Bioscience). One hour before measurement, the cell culture media was replaced with DMEM (Sigma-Aldrich) and 5 mM glucose adjusted to pH 7.4. Basal OCR was measured before the successive addition of 10 µM SYK-i2/BAY 61-3606, 1 µM isoproterenol (for inhibitor experiments) or the addition of 1 µM isoproterenol (for knock down experiments). Results are shown as average OCR + SEM of 18 measurements per time point and representative of three independent experiments.

**Glycerol release.** Following addition of fresh medium, cells were pretreated with SYK inhibitor and stimulated with ISO. Cell culture medium was collected after 6 h (brown) or 24 h (white) of stimulation and stored at −20 °C. Glycerol release was measured using the Adipolysis Assay Kit (Cayman Chemical, 10009381) following the instructions of the manufacturer.

**High fat diet feeding.** Male mice derived from multiple litters and born within one week were separated 21 days after birth into groups of two mice per cage and a small piece of the ear was removed from the left or right side to identify the mice and the tissue was used for genotyping. After one week of acclimatization, high fat diet (Research Diets Inc. #D12331) feeding started and the food was replaced weekly. Body weight was measured once per week with the investigators blinded to the genotype of the mice. Control cohorts were maintained on a chow diet (Labdiet, 5P00 Prolab RMH 3000). $F$ tests to compare variances were used to determine that the weight distribution of the AdipoQCre Syk$^{flox/flox}$ mice was significantly greater than in Syk$^{flox/flox}$ controls.

**Glucose and insulin tolerance tests.** For glucose tolerance test, mice were fasted for 12 h and injected intraperitoneally with 1 g/kg body weight glucose in sterile filtered 0.9% NaCl solution. For insulin tolerance tests, random fed mice were injected intraperitoneally with 0.75U/body weight pharmaceutical grade insulin (Humulin R, Lilly #0002-8215-01 (HI-210)). Blood glucose from the tail vein blood was measured using Contour blood glucose meter and strips (Bayer), again with the investigator blinded to the genotype of the mice. $F$ tests to compare variances were used to determine that glucose concentrations after 120 and 150 min were significantly greater in AdipoQCre Syk$^{flox/flox}$ mice than in Syk$^{flox/flox}$ control mice in glucose tolerance tests.

**Energy expenditure.** Oxygen consumption ($VO_2$) and carbon dioxide production ($VCO_2$) were measured by indirect gas calorimetry every 30 min or every 3 min for acute measurements using the metabolic chamber system from TSE Systems GmbH equipped with an optical beam activity monitoring device to measure movement of the mice. Age-matched and gender-matched mice were placed individually into sealed chambers supplied with chow diet and water under a 12 h light/12 h dark cycle at 20 °C housing temperature.

**Cold exposure.** Mice on a chow diet and housed at 20 °C were transferred to a cold environment (4 °C) in cages with 3–5 animals without food restriction (20 °C). The rectal core body temperature was measured using a microprobe thermometer from Physitemp (Model BAT-12).

**Genotyping.** Tail, BAT, WAT and cultured adipocyte DNA was isolated using the DNeasy Kit (Qiagen), amplified using Platinum PCR mastermix reagents (LifeTechnologies) and separated by a 1.5% agarose gel.
The primer pairs and PCR product sizes are given in Supplementary Table 2.

**Southern blotting.** 10 µg genomic BAT DNA were digested with 200 U BamHI (#R0136M, NEB) for 16 h at 37 °C and separated with a 0.7% agarose gel for 24 h at 80 V at 4 °C. Following acid-base hydrolysis, the DNA was transferred to charged nylon membranes (HYBOND-XL, #45001151, Fisher Scientific) for 12 h. 100 ng DNA probe (mouse Syk probe C[17], same as in Saijo et al.[19]) were labeled with 50 µCi dCTP-α−32P (BLU013H, Perkin Elmer) using a Prime-It II Random Primer Labeling Kit (#300385, Agilent). The labeled probe was purified with CHROMA SPIN + STE-30 Columns (#636058, Clontech) and hybridized with the membranes for 12 h at 60 °C. After washing, the membranes were exposed to autoradiography film (Biomax MS, #IB-8294985, VWR) for 24 h at −80 °C in the presence of an intensifying screen (#IB8518706, VWR).

**Quantitative PCR.** Total RNA was isolated from tissues or cells using TRizol or QIazol reagent (LifeTechnologies/Ambion) and a miRNAeasy kit (Qiagen). 300 ng were reverse transcribed using Superscript II reverse transcriptase (LifeTechnologies/Invitrogen) using random hexamers (LifeTechnologies/Invitrogen). The cDNA was diluted 1:10 and 2.5 µl for a 96-well plate or 1 µl for a 384 well plate were used for quantitative Real-time PCR. qPCR was carried out on an ABI7900HT Fast real-time PCR system (Applied Biosystems) and analyzed using the delta delta Ct method normalized to 18S. For RNA analysis of immortalized brown adipocytes a slightly different protocol was used: Total RNA was purified using TRI Reagent (Sigma-Aldrich). One µg of RNA was treated with RQ1 DNase (Promega) and reverse transcribed using Moloney murine leukemia virus reverse transcriptase (Life Technologies) using random hexamers (Bioline). cDNA was diluted 3 times in water. Diluted cDNA was analyzed by RT-qPCR using the Stratagene Mx3000P QPCR system. Each PCR mixture contained 1.5 µl cDNA, 10 ul SensiFast SYBR Lo-ROX kit (Bioline) and 2 pmol of each primer. The PCR was carried out in 96-well plates and results analyzed using the delta delta Ct method normalized to Tbp or 18S rRNA expression. Results are shown as pooled data from 3 to 4 independent experiments displaying the average and SEM. Primer sequences are listed in Supplementary Table 3.

**mRNA sequencing analysis.** Poly A+ RNA sequencing (TrueSeqStrandedPolyA) was performed on RNA samples from in vitro 4-hydroxy tamoxifen (4-OHT) treated day 8 brown adipocytes derived from CreERT2 Syk$^{flox/flox}$ and Syk$^{flox/flox}$ mice 6 h following stimulation with 0.1 mM isoproterenol in serum free medium and from unstimulated controls using a HiSeq genome sequencer (Illumina). RNA-Seq paired-end reads from Illumina 1.5 encoding were aligned using TopHat (v 2.0.13)[33] to the mouse genome (mm9; NCBI Build 37) with Ensembl annotation (release 67) in gtf format. mRNA sequencing was performed on three independent experiments. Differential analysis and counts per millions (cpm) were obtained by using HTSeq Count (intersection-strict mode) and edgeR[34]. Since the three replicates were done at different timepoints, we accounted for batch effects in the design matrix and model for the comparison in edgeR. Differential fold change used for identifying targets of $Syk$ deletion were calculated as follows. First, we used edgeR to calculate fold change before and after stimulation in both Syk$^{flox/flox}$ controls (referred to as FC (WT)) and CreERT2 Syk$^{flox/flox}$ (referred to as FC (KO)). We then identified genes with a difference in fold change i.e FC (WT)/ FC (KO) > 1.5 at $p$ value < 0.05. These genes are shown in red in Fig. 2b and were used for identifying GO categories (Fig. 2c). Enrichment analysis was performed by GSEA[35, 36] using the MSigDB (C2: curated gene sets) for genes identified in the comparison of the fold change of stimulated and unstimulated Syk$^{flox/flox}$ controls vs. stimulated and unstimulated CreERT2 Syk$^{flox/flox}$ brown adipocytes with a partial loss of SYK protein on day 8 of differentiation. Calculated fold changes and GSEA enrichment are listed in Supplementary Data 1.

**Immunoblotting.** Lysates were centrifuged for 10 min at 13,000 $g$ and 4 °C to remove debris, and NuPAGE sample buffer and reducing buffer (LifeTechnologies) were added after measuring and adjusting the samples for protein concentration (DC protein assay kit II, Bio-Rad). 2–20 µg protein per sample were separated for 2–4 h at 60–100 V using 8% 26 well NuPAGE Bis-Tris Midi gels in MOPS or MES buffer (LifeTechnologies) and in Criterion cells (Bio-Rad) using respective adapters (LifeTechnologies). Protein was wet transferred to polyvinylidene fluoride (PVDF) membranes (Immobilon P, Millipore) in Criterion blotter cells (Bio-Rad) using 2 x NuPAGE Transfer buffer (LifeTechnologies) with 10 % methanol for 25 min at 1 A. After blocking the membrane in filtered (Nalgene # 595–4520) TBS-T (50 mM Trisbase, 150 mM NaCl, 0.1% Tween 20 (Sigma-Aldrich # P1379)) with 3 % bovine serum albumin (BSA, Sigma-Aldrich # A7906) for 1 h, blots were incubated with primary antibody diluted in TBS-T 3% BSA sealed in hybridization bags and gently

shaking at 4 °C overnight, then washed three times for 10 min in TBS-T, incubated with secondary antibody (Cell Signaling Technologies) diluted in TBS-T 3% BSA gently shaking for one hour at room temperature, and then washed again three times for 10 min in TBS-T. Antibody binding was visualized using ECL Plus Western Lightning reagent (Perkin Elmer NEL 102) and blots were exposed to film (Kodak BioMax MR Film, Carestream Health Inc # 8701302). Films were scanned without adjustments using an Epson scanner. For immunoblotting of lysates from immortalized brown adipocytes, a slightly different protocol was used: 30 µg protein lysate was diluted in a buffer containing 2.5% SDS and 10% glycerol and was separated for 2.5 h using 4–12% Bis-Tris gradient gels (Life Technologies). Proteins were transferred to Immobilon PVDF membranes (Millipore) using 1X NuPAGE transfer buffer (Life Technologies) with 10% absolute ethanol. Membranes were blocked in TBST-T containing 5% nonfat dry milk or BSA (Sigma-Aldrich). Antibodies were diluted in TBST-T containing 5% nonfat dry milk or BSA and membranes were incubated for 2 h with primary and 1 h with secondary antibody with a washing step of $3 \times 10$ min before and after incubation with secondary antibody. The membranes were developed using enhanced chemiluminescense (ECL, Biological Industries) and the Fusion Fx (Vilber Lourmat) used for photodetection. All immunoblotting data shown were reproduced with almost identical results in at least one and typically two to three additional independent experiments. Experiments blotting for protein expression in ex vivo mouse tissues show lysates derived from individual mice in each lane. For the most important western blots, uncropped versions are provided in the Supplementary Figs. Antibody target antigen, clone and/or order number, manufacturer and application is provided in Supplementary Table 4.

**Histology**. To prepare formalin fixed sections, each freshly excised tissue sample of BAT, scWAT or visWAT was fixed in 15 ml 10% neutral buffered formalin solution (Sigma-Aldrich #HT5014) gently agitating for at least 24 h. After embedding the tissues in paraffin, 5 µm sections were prepared and stained with hematoxylin and eosin following established protocols.

**Oil red O staining of brown adipocyte cultures**. Cells were washed in 1X PBS and fixed in 3.7% formaldehyde solution for 1 h, followed by staining with Oil Red O for 1 h. Oil Red O was prepared by diluting a stock solution (0.5 g of Oil Red O (Sigma) in 100 ml of isopropanol) with water (6:4) followed by filtration. After staining, plates were washed twice with water and photographed using an iPhone.

**Immunofluorescence**. Cells isolated and grown as described above were seeded after the second split in 8 well culture slides (BioCoat, Corning, #354631) at a density of $1 \times 10^5$ per one well of an 8 well culture slide. Media was changed the next day and when generating knockout cells using the pre-adipocytes from CreERT2 Syk$^{flox/flox}$, 4-hydroxy tamoxifen (4-OHT) was added. After 24 h cells were induced to differentiate as described above and to generate KO cells in the presence of 4-OHT. Cells grown in 8 well culture slides were fixed for 20 min in 4% paraformaldehyde PBS solution. The cells were permeabilized with 0.25% tritonX 100 (Sigma, #T8787) for 10 min and washed 3 times with 1X PBS. Cells were then blocked for 30 min in 1× PBS with 3% BSA following incubation with the primary antibody (pSYK) in 1X PBS with 3% BSA for 45–60 min, washed three times with 1× PBS, and incubated with the secondary antibody, anti-rabbit F(ab)₂ IgG Alexa Fluor 488 for 45–60 min and washed three times with 1× PBS. DAPI counterstain was performed with 100 ng/ml DAPI for 10 min and washed for three times with 1× PBS. The top of the eight-well culture slides was removed and 15 µl of ProLong Gold antifading reagent (Invitrogen, # P36930) and a cover slip were applied. Pictures were taken with a Zeiss LSM 700 laser scanning confocal microscope and a 20x or 63x objective using a resolution of 2048 × 2048. Lasers were calibrated to achieve best signal to noise ration in the beginning and was not changed while taking pictures and changing experimental conditions.

**Light microscopy**. A bright field Zeiss AxioPlan2 upright microscope connected to a CCD camera with QCapture software was used to image all tissue sections stained with H&E and ORO.

**Statistical methods**. All statistical analyses were performed using Prism 7 (GraphPad Software, Inc). Statistical significance was indicated as follows: $*p < 0.05$; $**p < 0.005$; $***p < 0.0005$, and determined multiple hypothesis testing using the Holm-Sidak method by unpaired two-tailed $t$ test unless otherwise indicated. The $F$ test to compare variances was used to compare variances, and found to be similar between groups unless indicated in the respective method sections.

**Data availability**. Raw data generated during the RNA-seq procedure has been deposited in Gene Expression Omnibus (GEO) under accession code GSE91081. A complete list (excel file) of genes modulated in WT and KO cells is provided with the paper.

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

## Acknowledgements

We thank the imaging, bioinformatics, and genome core facilities at the Whitehead and the Koch Institute, the Dana-Farber/Harvard Cancer Center Specialized Histopathology Core (supported by NIH 5 P30 CA06516). We thank Daniel Miller for assistance with immunofluorescence and Alexander Tarakhovsky, Evan Rosen, William Comb, and Ronald Kahn for assistance or providing reagents. We offer special thanks to Heide Christine Patterson for helping perform and design experiments, in particular, western blotting, immunofluorescence, in vivo experiments using the Syk inhibitor, measuring oxygen consumption and rectal temperature of mice. We thank Dr. Patterson also for her help analyzing certain experiments, figure preparation and for help writing the manuscript. This research was supported by fellowship Kn1106/1-1 from the Deutsche Forschungsgemeinschaft DFG (M.K.), the EU FP7 project DIABAT (HEALTH-F2-2011-278373) (J.B.H.), and by RO1 DK047618 (H.F.L.).

## Author contributions

M.K. and S.W., performed the experiments, assisted and supported by A.N. (computational analysis and real-time PCR calculations), A.S. (tissue slides and H&E staining), T.E.C. (all in vivo experiments), H.Y. (Southern blot), M.J. (immunofluorescence), P.T. (computational analysis of RNA-seq), D.W.L (mouse protocol setup and initial experiments), and L.S. (establishing brown fat culture and initial experiments). J.B.H. and H.F.L. supervised and mentored the research. M.K., S.W., J.B.H., and H.F.L. wrote the manuscript with comments and contributions from all authors.

## Additional information

**Competing interests:** The authors have no financial conflict of interest.

