## [Peer Review File · Nature Communications]

Reviewers' Comments:

Reviewer #2:

Remarks to the Author:

In the updated manuscript, the authors addressed concerns raised during the review process. In particular, the involvement of subcutaneous fat was investigated and other thermogenic stimuli were used. Even though the *in vivo* proliferation and differentiation (brown fat formation) was not studied experimentally, the authors did provide reasonable explanation.

the overall conclusion is now greatly strengthened.

Reviewer #3:

Remarks to the Author:

The manuscript by Knoll et al., identifies SYK kinase as a regulator of brown adipocytes. The major strength of the paper is that SYK kinase has not been previously implicated in brown adipocytes, and it appears to have many roles in differentiation and in function. The major weakness of the paper is that the exact mechanism of SYK action in brown fat is not well defined, and its role in any one of the processes that it has been implicated in here has not been teased out, and so the rationale for the statement (LINE 40) that modulating SYK activity to treat diabetes is not that well supported by these data especially since many of the major signaling pathways are apparently affected by SYK. The authors have made some interesting findings and have addressed many of the concerns. But, there are a couple issues that need some clarification.

Firstly, I am confused by the statement (LINE 184) that the authors must confirm Syk deletion in the BAT progenitors of their AdipoQCre Syk flox/flox mice. This Cre is expressed in mature adipocytes, not precursors, and so this needs clarification as to what the authors mean by this. The CreERT2 system used later expresses everywhere (mature cells, precursors, and all other cells), but this did not have an effect on the BAT. Can this also be clearly explained?

I also don't quite understand the statement (LINE 59) that phosphorylated Syk is below the detection limit by Western blot. It seems to be quite abundant by IF. Does the pSyk signal also disappear in Syk KO cells (to confirm specificity)? Why does the Syk inhibitor block the upstream phosphorylation of Syk? What is the evidence that Syk is activated by isoproterenol-stimulated phosphorylation (Line 58)? For example are there changes in the phosphorylation of Syk substrates in response to stimulus/inhibition that might be predictive of its function in brown adipocytes?

Point-by-point response:

Reviewer #2 (Remarks to the Author):

In the updated manuscript, the authors addressed concerns raised during the review process. In particular, the involvement of subcutaneous fat was investigated and other thermogenic stimuli were used. Even though the in vivo proliferation and differentiation (brown fat formation) was not studied experimentally, the authors did provide a reasonable explanation. The overall conclusion is now greatly strengthened.

We thank the reviewer for the positive assessment of our manuscript.

Reviewer #3 (Remarks to the Author):

The manuscript by Knoll et al., identifies SYK kinase as a regulator of brown adipocytes. The major strength of the paper is that SYK kinase has not been previously implicated in brown adipocytes, and it appears to have many roles in differentiation and in function. The major weakness of the paper is that the exact mechanism of SYK action in brown fat is not well defined, and its role in any one of the processes that it has been implicated in here has not been teased out, and so the rationale for the statement (LINE 40) that modulating SYK activity to treat diabetes is not that well supported by these data especially since many of the major signaling pathways are apparently affected by SYK. The authors have made some interesting findings and have addressed many of the concerns. But, there are a couple issues that need some clarification.

We changed the manuscript Line 40 (now line 65) according to the reviewer's suggestion. The sentence now reads:

"These results establish SYK as an essential mediator of brown fat formation and function, and suggest that pharmacological modulation of SYK activity could have an impact on certain metabolic diseases."

Firstly, I am confused by the statement (LINE 184) that the authors must confirm Syk deletion in the BAT progenitors of their AdipoQCre Syk flox/flox mice. This Cre is expressed in mature adipocytes, not precursors, and so this needs clarification as to what the authors mean by this. The CreERT2 system used later expresses everywhere (mature cells, precursors, and all other cells), but this did not have an effect on the BAT. Can this also be clearly explained?

We thank the reviewer for pointing out this very important point, and we have rewritten the text to deal with it. We detected a small amount of activity of Cre recombinase in brown preadipocytes isolated from the BAT depots of AdipoQCre Syk^{flox/flox} and AdipoQCre Syk^{+flox} mice, as measured by PCR amplification of the deleted allele in the day 0 cells depicted in Figure 4g. After induction of these progenitor cells to differentiate in culture, the intensity of the deleted allele "band" increased, consistent with the known induction of adiponectin expression during BAT differentiation and thus with the induction of AdipoQ driven Cre expression. It was our mistake to state in the text that "adiponectin is expressed in preadipocytes." We have rephrased the text carefully; it now reads (line 216):

"To confirm the deletion of the Syk gene in BAT of AdipoQCre Syk^{flox/flox} mice, we isolated preadipocytes from BAT depots of AdipoQCre Syk^{flox/flox} and Syk^{flox/flox} control mice as well

from heterozygous $Syk^{+/flox}$ and AdipoQCre $Syk^{+/flox}$ mice and induced them to differentiate in vitro. We were able to detect faint amounts of the deleted allele at day 0, before induction of differentiation in progenitors from the AdipoQCre $Syk^{flox/flox}$ and AdipoQCre $Syk^{+/flox}$ mice but not the control mice, indicating a low level of expression of the AdipoQCre transgene and Cre activity in the progenitors (Fig. 4g). During differentiation of these AdipoQCre expressing progenitor cells in culture the intensity of the deleted allele “band” increased, correlating with induction of AdipoQCre expression during BAT differentiation (Fig. 4g). Thus the absence of deleted Syk in mature BAT cells from AdipoQCre $Syk^{flox/flox}$ mice is not due to faulty Cre recombinase activity, but rather to poor differentiation of Syk-deleted BAT progenitors”

Contrary to the reviewer’s assertion, and as already stated in the text, the CreERT2 system did have an effect on BAT. We induced Cre by treating the CreERT2 $Syk^{flox/flox}$ mice and $Syk^{flox/flox}$ control mice for 5 days with tamoxifen and measured oxygen consumption 16 days later. Syk indeed was deleted in BAT from the CreERT2 $Syk^{flox/flox}$ mice, but because these fat cells are presumably turning over slowly if at all, this loss of Syk did not affect the basal properties of these cells. As we stated in the text, “SYK is thus not required for survival of established BAT or maintenance of expression of brown fat characteristic genes” Consistent with our cell culture experiments showing that Syk is essential for beta-adrenergic increases in BAT gene expression and oxygen consumption, thermogenesis in these CreERT2 $Syk^{flox/flox}$ mice induced by the β 3-specific agonist CL316, 243 was greatly reduced. As already stated in the text, “However, as judged by measuring the oxygen consumption of these mice in metabolic chambers, CreERT2 $Syk^{flox/flox}$ mice failed to induce CL 316,243-induced oxygen consumption, as contrasted with control $Syk^{flox/flox}$ mice (Fig. 4l). SYK signaling thus promotes β -adrenergic induced BAT activation in vivo.” Thus we feel it is unnecessary to make any changes to this segment of the text.

I also don't quite understand the statement (LINE 59) that phosphorylated Syk is below the detection limit by Western blot. It seems to be quite abundant by IF. Does the pSyk signal also disappear in Syk KO cells (to confirm specificity)?

We thank the reviewer for this important point, but many antibodies can recognize their target by immunofluorescence and not by Western blotting, and vice versa. For Western blotting the proteins are denatured and transferred from gels to filters. In immunofluorescence studies the cells are cross-linked but generally it is the native protein that is detected. Further, we have added more IF data to Supplementary Fig. 1a with a lower magnification showing that not all cells show an increase of pSYK in response to isoproterenol which also could be the reason that we don’t detect a signal in western blot. We carefully rephrased the sentence to read (now line 80):

“Although we were unable to detect phosphorylated SYK by western blotting, we detected an increase of phosphorylated SYK following isoproterenol treatment using immunofluorescence (IF) that was blocked when the cells were pretreated with SYK inhibitor R406 (SYK-i3) (Fig. 1d)”

To test the specificity of the pSYK antibody we now added new IF data performed on KO cells derived from CreERT2 mice from three independent experiments as shown in figure 1f and 2a with a 90% reduction of SYK protein. In parallel we repeated the experiment from Figure 1d from three independent experiments to confirm isoproterenol induced SYK phosphorylation in WT cells. We do see an increase of the pSYK signal in isoproterenol stimulated WT cells (Supplementary Fig. 1a) that was blocked when the cells were pretreated with SYK-i3. Similarly, no phosphorylated SYK signal was detected in isoproterenol

stimulated KO cells (Supplementary Fig. 1a). Thus we are confident that the pSYK antibody used in this study is specific.

Why does the Syk inhibitor block the upstream phosphorylation of Syk?

The R406 Syk inhibitor is an ATP competitive inhibitor and that's why this question is very reasonable. Syk is known to autophosphorylate its linker region to sustain activation, hence R406 might inhibit this autophosphorylation. We used the pSyk 323 antibody that detects phosphorylation in the linker region. We don't know if the phosphorylation of other tyrosines is blocked because other pSyk antibodies we tested didn't work well in our experiments.

What is the evidence that Syk is activated by isoproterenol-stimulated phosphorylation (Line 58)? For example are there changes in the phosphorylation of Syk substrates in response to stimulus/inhibition that might be predictive of its function in brown adipocytes?

This is a very important point and also very difficult to answer. First, it is known from the literature that Syk activation is regulated by phosphorylation. We realize that line 58 was misplaced as it relates to our IF findings described after the statement. It has now been updated to read (line 79): "SYK activity is controlled by phosphorylation."

We do see an increase of Syk phosphorylation after stimulation with isoproterenol using IF and this is indicative of activation of Syk. In both B cells and mouse fibroblasts hundreds of proteins become tyrosine phosphorylated after Syk induction and their phosphorylation is reduced in the presence of Syk inhibitors or Syk knock down. However, it is very difficult to discern which of these are direct substrates of Syk and which are phosphorylated by kinases downstream of Syk. We did test the phosphorylation status of the Syk substrate Btk after isoproterenol stimulation and it was unchanged. Thus at this point we are unable to identify the proteins directly phosphorylated by Syk after isoproterenol stimulation and this is part of our current efforts in the laboratory.